# Prevalence of Key Modifiable Cardiovascular Risk Factors among Urban Adolescents: The CRO-PALS Study

**DOI:** 10.3390/ijerph17093162

**Published:** 2020-05-02

**Authors:** Ivan Radman, Maroje Sorić, Marjeta Mišigoj-Duraković

**Affiliations:** 1Faculty of Kinesiology, University of Zagreb, 10000 Zagreb, Croatia; maroje.soric@kif.unizg.hr (M.S.); marjeta.misigoj-durakovic@kif.unizg.hr (M.M.-D.); 2Faculty of Sport, University of Ljubljana, 1000 Ljubljana, Slovenia

**Keywords:** teenage, metropolitan, high school, lifestyle, unhealthy behavior

## Abstract

The occurrence of chronic diseases in youth has become a serious global issue requiring early prevention. Due to the specific environment in large cities, urban youth are especially exposed to risky lifestyle patterns. Objective: This survey aimed to determine the prevalence of key modifiable cardiovascular risk factors in the adolescent population of the Croatian capital Zagreb. Methods: A clustered two-stage random sample design was employed to select a representative group of 903 adolescents (girls *n* = 444; boys *n* = 459; age 15.6 ± 0.4) starting their secondary education. Participants were invited to fulfil an electronic questionnaire meant to collect data on daily physical activity, screen time and tobacco use. In addition, anthropometric and blood pressure measurements were taken by medically trained personnel. Data that were not originally categorical were dichotomized based on internationally accepted cut-off values for each risk factor, summarized for females and males, and presented as percentages and 95% confidence intervals. Results: The outcomes suggest that more than one half did not meet the recommended daily physical activity (girls 59.4%; boys 45.5%), a vast majority exceeded 2 h of screen time per day (girls 87.9%; boys 91.6%), and one quarter had high blood pressure (girls 21.6%; boys 27.0%). Moreover, the results indicated nearly one fifth of adolescents had excess weight (girls 16.1%; boys 22.2%) and a similar proportion smoked tobacco (girls 20.8%; boys 17.0%). Conclusions: Local and regional health stakeholders should make additional efforts to promote healthy lifestyles in urban teenagers. Special emphasis should be placed on promoting physical activity.

## 1. Introduction

During the last decades, cardiovascular diseases (CVDs) have become the leading global source of mortality, estimated to cause 31.5% of all deaths and 45% of all non-communicable disease (NCD) deaths each year [1]. Despite the fact that mortality due to CVD has decreased in Europe [2,3,4], CVD remains the most common source of death in Europeans [5]. Recent epidemiological updates and statistics suggest that CVD causes around 4 million deaths in Europe and more than 1.8 million deaths in the European Union on an annual basis [5,6], accounting for 45% and 37% of all deaths in Europe and the European Union, respectively.

Modifiable behavioral patterns such as physical inactivity, sedentary time, and tobacco use are recognized among the small group of factors increasing the risk of developing physiologic conditions such as excess weight and high blood pressure, both known to underly most CVDs (WHO 2019). Recent studies suggest that insufficient physical activity and high sitting time cause more than 7.3% of all-cause deaths across Europe [7] and about 3.8% of all-cause deaths worldwide [8]. The estimated annual rate of all-cause mortality attributable to tobacco use totals approximately 16.0%, being more than three times higher in men than in women [9]. In addition, annual CVD-related deaths attributable to excess weight in the European Union amount to nearly 17.3% [10], while high blood pressure is estimated to cause nearly 12.8% of all-cause deaths worldwide. 

Recently, an increased frequency of chronic diseases in youth and adolescence has been reported [11,12]. This should not be surprising given that modern lifestyle and the increased incidence of risk factors in young people favor the development of chronic diseases [13,14,15]. It is not unusual that life habits acquired in the earlier stages of life persist throughout adulthood, [16,17], and therefore these groups are at high risk of acquiring chronic diseases.

Urban areas, such as large cities or capitals, are somewhat more susceptible to risk factors due to the specific environmental and lifestyle factors [18,19]. On the European continent, this is vastly true for the countries in rapid transition in central, southern, and eastern Europe, which are economically lagging behind the countries of western Europe [20]. In these countries, the capitals often account for a larger share of the total population. According to the available data, the national campaigns and prevention measures fail to adequately forestall emerging secular trends in urban centers such as poor diet, physical inactivity, and a sedentary lifestyle.

In large cities there might be a trend in health behavior patterns because of the high population density and general tendency of people to form socially cohesive groups in urban spaces [21]. This seems particularly common among young people due to regular school attendance and age-specific peer influence [22,23]. Adopting common patterns of behavior in areas of poor health policies increases the likelihood of adopting a disease-leading lifestyle. In order to create prevention policies that may contribute to reducing the incidence of risk factors in transition countries, it is important to gain insight into the frequency of occurrence of particular risk factors. While the occurrence of health risks in Croatian adult and general youth populations has been investigated [24,25], published data on the prevalence of health risks in the urban adolescent population seem scarce. 

Hence, the goal of this investigation was to determine the frequency and sex distribution of cardiovascular risk factors in the urban adolescent population of the Croatian capital. Zagreb is a city that in its size, ethnic, climate, infrastructure, and economic characteristics does not significantly vary from most cities of central and south-eastern Europe. The findings of this study may contribute to the formulation of risk-management policies for adolescents in urban areas of central and south-eastern Europe. 

## 2. Materials and Methods 

### 2.1. Study Participants 

This report is based on data collected during the first of the four-year-long observational CRO-PALS (Croatian Physical Activity in Adolescence Longitudinal Study) study. The study procedures were designed to explore health-related habits and risk behavior among the adolescents in the city of Zagreb (Croatia, EU) during the high-school period and were thoroughly described elsewhere [26]. Briefly, a clustered two-stage random sample design was employed to select a representative group of students starting their secondary (high school) education. In the first stage, all high schools in the city of Zagreb (n = 86) were clustered by type as “grammar”, “vocational”, and “private”. Based on the original share of school types and average number of registered students per school, 13 public (eight vocational and five grammar schools) and one private school (grammar school) were selected. In the second stage, half of all first-grade classes in each school were randomly selected, and a total of 1408 students received an invitation to participate. Written parental consent for involvement in the study was obtained for 903 students (response rate = 64%). All students and their parents signed an informed consent form in accordance with the Declaration of Helsinki. The study was approved by the Institutional Review Board of the Faculty of Kinesiology, University of Zagreb, Croatia, under No. 1009-2014.

### 2.2. Measurements

Anthropometric measurements including body weight, body height, and waist circumference were performed by trained medical personnel. Participants were barefoot and lightly clothed for all measurements. Body weight was measured to the nearest 0.1 kg using a portable medical balance scale, body height was measured to the nearest 0.1 cm using a GPM anthropometer (Siber-Hegner & Co., Zurich, Switzerland), and waist circumference was measured to the nearest 0.1 cm using a standard measurement tape. Subsequently, the body mass index (BMI) was calculated as kg/m^2^ for each participant and was rated according to the age- and sex-specific International Obesity Task Force (IOTF) cut-off values [27]. In addition, each participants’ waist circumference was compared to the criterion to determine individual levels of abdominal obesity. Based on the cut-off values of the 90th and 97th centiles for German adolescents of respective age between 2003–2007 [28], participants were allocated to three categories: <90th percentile, 90th–97th percentile, and >97th percentile.

An electronic form of the questionnaire was used to collect lifestyle-related data and demographic information. First, to assess compliance to the global health guidelines on the amount of physical activity, the School Health Action, Planning, and Evaluation System (SHAPES) questionnaire was used [29]. The measurement properties of the used method are similar to other recall questionnaires used to assess physical activity in adolescents [29]. The questionnaire requires students to recall the number of hours and minutes in 15-min increments that moderate physical activity (MPA) and vigorous physical activity (VPA) were performed each day of the previous seven days. Daily duration of MPA and VPA time was summed for each day to assess matching to the guidelines. Participants that failed to reach the recommended minimum on each of seven reported days were classified as “not reaching the minimum of 60 min of MVPA per day”. Second, the SHAPES questionnaire was further applied to assess screen time through the items requiring participants to report the daily time spent playing computer games, viewing television, and browsing the Internet for the past week. The response form indicated the number of hours and minutes in 15-min increments spent on particular behaviors [29]. Time spent on the three behaviors was summed for each day, and the mean daily screen time was calculated. Participants were divided into two categories according to the cut-off value of an average of 120 min of screen time per day. In addition to physically active and passive daily time self-reporting, a described electronic form of the prevalence of smoking was evaluated with the multiple response question “Do you smoke cigarettes?”. This required students to elect one of three possible responses: yes, no or only occasionally. Finally, the socioeconomic status of the students was assessed based on the question “How wealthy do you think you are, compared to your peers?”. The responses were arranged along a Likert 5-point scale: 1 = substantially above average, 2 = slightly above average, 3 = average, 4 = slightly below average, and 5 = substantially below average. 

Blood pressure measurements were taken by three medically trained research team members following standard guidelines of the European Society of Hypertension [30]. In short, students were screened by an auscultatory method using a mercury sphygmomanometer to obtain two blood pressure readings in 1-min intervals after at least 5-min of rest in a sitting position. BP was measured on the right arm using an appropriately sized cuff. To avoid terminal digit preference, blood pressure was recorded to the nearest 2 mmHg. The average value of two readings was used for analyses. In the case anxiety was suspected, another reading was taken after 5 min, during which investigators attempted to reassure participants and reduce anxiety. In these participants, only the last reading was used for analyses, unless it was the highest, in which case the average value was registered. High blood pressure was considered as systolic blood pressure and/or diastolic blood pressure that exceeded the 95th percentile for age, sex, and height [31].

### 2.3. Data Analysis

Collected data were processed using SPSS software, version 24.0 (IBM, New York, NY, USA). All analyzed variables were first tested for normality using the Kolmogorov–Smirnov test and by inspecting histograms and normal probability plots. To present the prevalence of risk behavior among participants stratified by sex, methods of descriptive statistics were applied. Demographic data are presented as the mean and standard deviation for continuous variables and as frequencies for categorical variables. Behavioral variables were all categorical and were presented graphically (i.e., histograms) as percentage values and 95% confidence intervals. The lower and upper confidence limits for the 95% confidence interval of percentages were generated by SPSS using the equal-tailed Jeffreys prior method.

## 3. Results

Descriptive parameters of the study participants (age and socioeconomic status) are displayed in Table 1. Boys and girls were equally represented, and the large majority of the participants perceived themselves as having average or below average wealth, in comparison to their peers. The incidence of each of four BMI categories based on IOTF cut-off values and abdominal obesity percentiles are depicted in Figure 1. The cumulative prevalence of overweight and obesity amounted to 22.2 (18.5–26.3)% and 16.1 (12.8–19.9)% for male and female adolescents, respectively, while abdominal obesity (waist circumference >90th percentile) was recorded in 10.5 (7.9–13.7)% males and 7.3 (5.1–10.1)% females. The reported tobacco use among urban adolescents suggested that 15.0 (11.8–18.7)% and 5.8 (3.9–8.4)% girls and 13.3 (10.3–16.7)% and 3.7 (2.2–5.8)% boys (first high-school graders) in the Croatian capital were habitual and occasional smokers, respectively. 

Results further showed that a major proportion of all female adolescents (59.4 (54.6–64.1)%) did not comply to current health guidelines of reaching a minimum of 60 min of MVPA per day (Figure 1). In the male population of adolescents, about one half met the recommended minimum. A vast majority of both boys and girls spent more than 120 min/day on average in front of screens (91.6 (88.7–94.0)% and 87.9 (84.5–90.8)%). Finally, 21.6 (17.9–25.8)% of females and about 27.0 (23.0–31.4)% of males in the urban adolescent population were identified as having blood pressure values higher than normal.

## 4. Discussion

The current study is based on data collected from a representative sample of urban adolescents, first high-school-graders, in Zagreb, the capital of Croatia, and one of larger urban zones in central Europe. Present findings demonstrate the occurrence of selected risk factors in urban novice high-school students and highlight their distribution across sexes. To the best of the authors; knowledge, this is the first study reporting the prevalence of CVD risk factors in the above population. It should be, however, noted that this report considers only key modifiable risk factors (excess weight, high blood pressure, insufficient physical activity, excess screen time, and smoking). The main findings suggest that more than one half did not meet recommended minimum of daily physical activity, nine tenths spent more than two hours in front of the screen per day, and about one quarter of observed adolescents had high blood pressure. Moreover, results indicated that nearly one fifth of adolescents had excess weight, and the same proportion habitually or occasionally smoked. The meaning of the above results within the context of current global public-health policies and recent regional findings as well as the shortcomings of current study are further discussed.

Behavioral patterns such as physical inactivity, sedentary behavior, and smoking are linked with muscle atrophy, insulin resistance, altered energy balance, blood vessel damage and narrowing, clot formation, and peripheral blood flow reductions; hence, they are perceived as immediate contributors to physiological risk factors such as a high BMI and high blood pressure [32,33,34,35]. Because of their modifiable nature and conscious/voluntary control, physical activity and screen time have recommended minimum and maximum values, respectively. Although global guidelines endorse generally increasing physical activity and reducing screen time, for maintaining health, a minimum of 60 min of MVPA and a maximum of 120 min of screen time daily have been recommended [36,37,38]. Due to the strong association of tobacco smoking with the development of CVDs and all-cause mortality, international and national policies seek to eradicate this behavior [39,40]. Recent global health reports suggest that nearly 81% (84% girls vs. 78% boys) and 56%–65% of adolescents worldwide do not meet the recommended minimum for PA and maximum for ST, respectively [41,42,43], while the global proportion of smokers among adolescents totals nearly 9.5%, with the highest prevalence of 19.2% in the European population [44]. Current results indicate a lower rate of insufficient physical activity (59.7% girls and 45.5% boys) and considerably higher rate of excess screen time (both sexes nearly 90%) in the studied population in comparison to worldwide reports. It seems that the adolescent population in Zagreb is less insufficiently active when compared to Budapest, the capital of the nearby Hungary, with 86.5% and 75.8% insufficiently active girls and boys [45], or when compared to Slovak urban zones including Bratislava, with 78.9–91.4% and 60.4–77.7% inactive girls and boys, respectively [46,47]. Conversely, the incidence of nearly 90% excessive screen time in the current study population was considerably higher than the 41% reported in 2015 for a similar population in the neighboring Slovenian capital Ljubljana [48]. The above comparisons should, however, be considered with certain caution. It is likely that at least a part of the observed difference may have arisen due to different instruments, recall times, and activity clarifications applied to assess physical activity and screen time across the studies. Specifically, for the reported prevalence of physical inactivity in Budapest, the assessment methodology remained rather unclear, while in both Slovak studies, the methodology was based on a general question asking to state how many times the exercise was performed for a reference period, which ranged from one to six months, therefore not allowing one to either quantify or evaluate the level of daily physical activity. In addition, inactivity was defined differently across studies; for instance, in the study of Pitel et al. [47], insufficient physical activity referred to an individual engaging in less than two days per week, while in the present study, inactivity referred to all individuals reporting less than 60 min for at least one day per week. In the current study, a recall period was one week, and the students were asked to quantify the exact amount of physical activity time for each particular day of the past week. Disregarding the assessment method used, it is worth noting that the girls in this study tended to meet the physical activity guidelines less than boys. This observation is in line with both the global and regional trends described above. 

As somewhat expected, the proportion of smokers among high-school students in the Croatian capital was higher than the global average, but the proportion remained comparable to the above-mentioned European percentage. In contrast to observed global trends [44], the current results indicated that girls in this study were more likely to smoke cigarettes, both habitually and occasionally, in comparison to boys. The prevalence of cigarette smoking in Zagreb adolescents is comparable to results reported for urban Slovak areas (23.5% in girls and 20.8% in boys [46,47]), notably higher in comparison to Ljubljana (girls 13.3%; boys 10.8%; [48]), and lower in comparison to Budapest (38% girls; 47% boys; [45]). Note, however, that above comparisons should be considered with certain caution due to inconsistencies in methodology. Specifically, the present study evaluated smoking by means of one multiple response question suggesting one to three possible answers, while some of the compared studies used either more precise quantifications of cigarettes per time unit or asked different multiple response questions.

Clinically determined conditions such as high BMI and raised blood pressure are among key physiological predictors of CVD development. The reported worldwide cumulative prevalence of overweight and obesity among children and adolescents aged 5–19 in 2016 exceeded 18%, with nearly equal proportions of boys and girls [49]. Except for the Czech Republic and Italian adolescents, it seems that nearly 22 to 25% of European adolescents were overweight or obese at that time [50]. Likewise, a relatively recent review and meta-analysis pointed out that pooled prevalence of high blood pressure for adolescent boys and girls totaled nearly 11.2% [51]. Among the current population of urban adolescents, the rate of those who are overweight plus obese is lower compared to both worldwide and Europewide incidences. In contrast to global prevalence, the distribution of excess weight in the current population suggests greater risk in male adolescents. Similar to the present investigation, a greater proportion of overweight and obese boys was reported for the last available generation of 14-year-olds in Ljubljana (23.5% boys vs. 14.2% girls; [52]) and the joint Bratislava and Kosice adolescent population (27.0% boys vs. 22.5% girls; [46]). A somewhat closer pooled rate and different distribution were recently observed for Budapest, where 15.3% of the boys and 19.0% of the girls were overweight and obese [45]. However, note that the prevalence of cumulative overweight and obese boys and girls in the present study (22.2% boys and 16.1% girls) was comparable to the results of the Slovenian study and lower than that reported for Slovakia. Regarding the prevalence of elevated blood pressure, identified prevalence in both current male (27.0%) and female (21.6%) adolescent populations doubled compared to global estimates for 2014 (13.0% in boys and 9.6% in girls; [48]). These results are to a certain extent expected, since the recent data identified Croatian men as having the highest prevalence of hypertension globally [53]. In the regional context, girls’ values are comparable to data observed in 14–18-year-old Budapest females (23.2%) but not male adolescents, which were reported to be double those observed in the present study (46%; [54]). In contrast, a notably lower rate of high blood pressure compared to present results was observed in neighboring Belgrade’s children aged 7–14, ranging between 5.6% in boys and 5.3% in girls [55]. This diversity in the results obtained in different studies can partly be attributed to differences in the methodology (i.e., setting, type of instrument used for measuring blood pressure, cuff size, number of measurements, etc.), age, and weight status of the study participants. 

Based on current epidemiological data, it is difficult to speculate the reasons behind the fact that urban youth males appear to be at a greater risk of being overweight and obese with high blood pressure. This is especially due to the results that a lower incidence of physical inactivity in Croatian younger men versus women was observed both in this study and previously [56]. Concerning the fact that the current male adolescents have consistently higher levels of physical activity and sport participation, as well as lower levels of sedentary time [57], a rational explanation might be that the higher dietary intake in the current male population [58] combined with generally poorer nutrition habits in Croatian young men [59] may have led to the greater overweight and obesity prevalence. In addition, the traditionally spicy food and increased salt intake in the Croatian population [59,60,61,62] and high prevalence of elevated BMI could have likely influenced the proportion of urban young males with elevated blood pressure [60,63]. 

The current study should be viewed in light of its limitations and strengths. The main weakness of this study is a recall method used to obtain information related to behavioral data, resulting in a possible recall bias. Besides, omitting poor dietary habits and alcohol misuse is another disadvantage concerning the context of health-risk incidences. However, note that the fairly inclusive electronic questionnaire enabled us to collect robust data on activity related behavioral risks from a highly representative group of urban adolescents, comprising school students registered at all offered school types. Although blood pressure measurements were taken repeatedly by an auscultatory method using a sphygmomanometer, all measurements were recorded on a single occasion, which may overrate the prevalence of elevated blood pressure. To minimize the phenomenon of white coat hypertension, blood pressure measurements were performed in school settings and by informally dressed medically trained research team members.

Taken together with the previous research, and according to the current study results and limitations, two implications for future research may be suggested. Epidemiological studies focusing Croatian urban adolescent populations should consider investigating the prevalence of poor nutrition habits and alcohol misuse. In addition, to increase general comparability of the reports focusing on health risks in youth living in urban areas, prospective surveys should strive to unify data collection methods.

## 5. Conclusions

The trend of the high occurrence of behavioral and physiological risk factors for CVD among adolescents is present in the area of the Croatian capital. The current population of urban adolescents is predominately exposed to the risk arising due to insufficient physical activity and excessive screen time. The incidence of cigarette smoking is comparable to most European sites, but still substantially higher compared to the recent global trends. The observed adolescents have a high risk of high blood pressure, as the prevalence is twice the global average. It seems that local and regional public health policies should particularly focus their efforts on the prevention and treatment of insufficient physical activity, cigarette smoking, and high blood pressure. 

## Figures and Tables

**Figure 1 ijerph-17-03162-f001:**
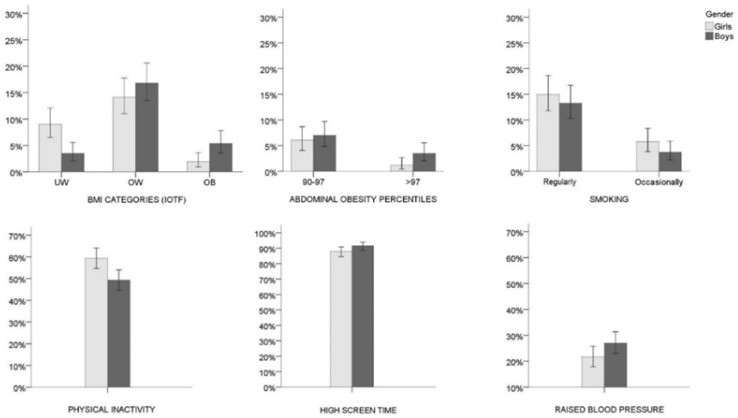
Prevalence of the body mass index (BMI) categories (girls *N* = 410; boys *N* = 428), abdominal obesity percentile categories (girls *N* = 411; boys *N* = 429), smoking (girls *N* = 414; boys *N* = 429), physical inactivity (girls *N* = 414; boys *N* = 430), high screen time (girls *N* = 414; boys *N* = 429), and raised blood pressure (girls *N* = 416; boys *N* = 429) stratified by sex. Error bars represent 95% confidence levels; IOTF = International Obesity Task Force; UW = underweight; OW = overweight; OB = obese. Note: Girls and boys were allocated to the BMI categories according to the IOTF cut-off values proposed by Cole and Lobstein, 2012 [27], and to the abdominal obesity categories based on cut-off values of 90th and 97th centiles for German adolescents of respective age between 2003–2007 (Kromeyer-Hauschild et al., 2010 [28]); physical inactivity refers to <60 min of moderate-to-vigorous physical activity per day; high screen time refers to >120 min of screen related activities per day; Raised blood pressure refers to systolic and/or diastolic blood pressure above the 95th percentile for age, sex, and height as described by Falkner et al., 2001 [31].

**Table 1 ijerph-17-03162-t001:** Basic characteristics of participants stratified by sex.

	Girls	Boys
*n*	444	459
Age (y)	15.6 ± 0.4 ^a^	15.7 ± 0.4 ^a^
BMI (kg/m^2^)	21.4 ± 3.1 ^a^	21.8 ± 3.6 ^a^
Waist circumference (cm)	69.2 ± 6.7 ^a^	75.1 ± 8.4 ^a^
Socioeconomic status (%)		
1—substantially above average	1.2	0.2
2—slightly above average	7.1	4.6
3—average	46.8	44.9
4—slightly below average	33.6	34.5
5—substantially below average	11.3	15.7

BMI = body mass index; ^a^ arithmetic mean ± standard deviation.

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
