# Peer review of "Prevalence of Key Modifiable Cardiovascular Risk Factors among Urban Adolescents: The CRO-PALS Study"

_ijerph, 2020, doi:10.3390/ijerph17093162_

Round 1

Reviewer 1 Report

Abstract

  • Results should not be stated in terms of inferences
  • What statistical tests were used?

Introduction

  • Lines 55-60-Not sure what the authors mean by "tendency to transmit lifestyle patterns..". There might be a trend in health behavior patterns but one cannot say that these are being transmitted. The authors should be careful in terms of how they phrase their sentences.
  • Also, there are no references to support lines 55 to 60.
  • While the authors give an overview of how young people are at risk to chronic conditions, they do not put the study in context. They do not mention what studies have been conducted in Croatia; whether there are research gaps and how this study fills in those gaps.

Other comments

  • Some major editing is required in terms of how the sentences are written and there are also some grammatical errors
  • Example-Line 31: ..risk behaviors...
  • E.g., line 15: ..clustered two-stage sample random design was employed
    Line 22: "have" instead of "having"

Results

  • To state the results in the past tense consistently.
  • What are the reasons that the authors decided to focus on only the selected modifiable risk factors? Why were other risk factors such as diet not assessed?
  • They need to address that as a limitation as well.

Discussion

  • While the authors compare their findings to other studies conducted in Europe, they rarely go in-depth on what are the plausible reasons behind their findings. Why did they find similar or different findings?
  • I suggest the authors also add a section on implications for future research-what are the next possible research steps/questions they recommend in light of their findings and other studies?

Reviewer 2 Report

The manuscript presents the interesting issue related to assessment of the frequency and sex distribution of  cardiovascular risk factors among adolescents in Zagreb.

I have made several suggestions that the Authors may want to consider in revising their manuscript.

INTRODUCTION

The hypotheses should be formulated and should be explained.

MATERIALS AND METHODS

It would be helpful to add the inclusion/exclusion criteria in the study.

DISCUSSION

THE DISCUSSION needs to go more in depth and the Authors need to be more critical about their own results. In addition, they should specify which of their hypotheses have been confirmed. How can be explained the meaning of all obtained results? A more detailed discussion in light of the existing research literature should be presented. The Author should explain  the results as well as give a more detailed outlook on the continuation of the study considering the results.

Reviewer 3 Report

All studies that are able to describe the actual state of life style and health in young populations are very actual. The data about life style of youth are often collected in many countries but the practical reccommendations are often to general. General reccommedatins that should be applicable in all countries do not exist. The main problem is connected with the experiences and long laxting development of concrete country. From this point of view, all nation studies my help  by the solving this problem over the world. Formally it is neessary to add more concrete details in the abstact a  in dis ussion more e plain the reasons that are responsible for results.

Round 2

Reviewer 1 Report

Thank you for making the changes previously requested. The manuscripts reads better. However, there are still some issues with understanding sentences and grammatical errors (example line 150: "amounted to"; line 77- behaviors should be in plural; 

Some parts of the results section are not in the past tense (for example, lines 161-167); same for the discussion when the authors are referring back to their findings.